# Analysis of Prevalence and Related Factors of Cyberbullying–Victimization among Adolescents

**DOI:** 10.3390/children11101193

**Published:** 2024-09-29

**Authors:** Jun Ma, Liyan Su, Minhui Li, Jiating Sheng, Fangdu Liu, Xujun Zhang, Yaming Yang, Yue Xiao

**Affiliations:** 1Department of Epidemiology and Biostatistics, School of Public Health, Southeast University, Nanjing 210009, China; 220223675@seu.edu.cn (J.M.); 220214008@seu.edu.cn (M.L.); 220213949@seu.edu.cn (J.S.);; 2Yixing Center for Disease Control and Prevention, Yixing 214200, Chinaxy.ybb@163.com (Y.X.)

**Keywords:** cyberbullying–victimization, adolescents, regression analysis

## Abstract

Background/Objectives: Cyberbullying is an increasingly serious issue that negatively impacts the mental and physical health of adolescents. This study aims to report the prevalence rates of adolescent cyberbullying–victimization and its associated related factors, providing a scientific basis for targeted efforts to protect the mental and physical well-being of adolescents; Methods: From March to May 2019, there were 13 high schools and 33 middle schools in Yixing, with a student ratio of 2:1 between middle and high school. Using a random cluster sampling method, we selected four high schools and three middle schools based on this ratio, resulting in a total of 13,258 students. We conducted a survey using a self-designed questionnaire to investigate the experiences of adolescents with cyberbullying and victimization, comparing the differences in cyberbullying–victimization based on various demographic characteristics. Additionally, we employed a multifactorial logistic regression model to analyze the associated factors; Results: The rate of adolescents who declared themselves as cyberbully-victims is 2.9%. The results of the logistic regression analysis indicate that being male, having both parents working outside the home, experiencing occasional or large conflicts among family members, being subjected to punishment-and-abuse child discipline, always or often using social software (websites), enjoying playing single or multiplayer games, self-smoking, and self-drinking were associated with a higher likelihood of being a cyberbully-victim (*p* < 0.05); Conclusions: Adolescent cyberbullying–victimization is affected by personal, family, and social factors. Therefore, comprehensive strategies and measures are needed to intervene in this problem.

## 1. Introduction

The rapid development of the Internet has facilitated many activities in our daily lives, supporting activities such as learning and communication. However, it has also led to negative phenomena, such as cyberbullying [1,2]. Cyberbullying refers to the deliberate and sustained use of electronic devices or other forms of digital communication technology to harm or attack others [3,4,5]. Approximately one-third of global Internet users are under the age of 18 [6,7], making them vulnerable to both experiencing and perpetrating cyberbullying [8,9,10]. Studies [11,12,13] have reported that the prevalence of cyberbullying among adolescents in China ranges from 31.4% to 59.0%. Similarly, international studies [2,14,15] have found that the prevalence among Canadian adolescents is between 38.0% and 48.0%, and in the United States it ranges from 13.9% to 57.5%. A 2023 systematic review [3] suggested that the COVID-19 pandemic may have influenced the prevalence of cyberbullying and cybervictimization among adolescents worldwide, with some studies reporting an increase in certain regions, such as Asia and Australia [16], while others observed a decline in Western countries [17]. These changes could be attributed to varying contextual factors, such as school closures and increased online activity during the pandemic.

Unlike traditional bullying, cyberbullying is both virtual and covert, posing significant threats to the physical and mental development of children and adolescents, and has become a critical public health concern [18,19]. Previous research on cyberbullying has predominantly focused on the perpetrators, with limited attention given to victims, particularly those who are both bullies and victims [2,20,21]. The underlying reasons for and influencing factors of this phenomenon have not been adequately discussed. This group simultaneously plays dual roles as both victims and perpetrators, leading to greater psychological stress and distress [22,23,24]. Therefore, studying cyberbullying–victimization can help elucidate the mechanisms behind such behaviors and provide a theoretical basis for developing more targeted intervention strategies.

This study is based on Bronfenbrenner’s [25] social ecological theory, which posits that cyberbullying is not a singular phenomenon but rather the result of multiple factors interacting, including individual characteristics, family environment, peer relationships, and school climate. Previous studies [26,27,28] have shown that factors such as gender, age, academic performance, and behavioral habits (e.g., smoking and drinking) are closely associated with the occurrence of bullying behavior. For example, male adolescents are more likely to become bullies than female adolescents [29,30,31,32], although females are more inclined to engage in indirect forms of bullying [29,30,31,32]. Furthermore, the family environment plays a crucial role in preventing bullying. Close relationships with parents, perceived emotional support, parental supervision and involvement, and the establishment of family rules are all considered important protective factors against both cyberbullying and traditional bullying [33,34,35,36]. At the school level, positive peer relationships and an inclusive school climate are also recognized as key factors in preventing bullying behavior [37,38]. The objective of this study is to examine the prevalence of cyberbullying and victimization, along with their associated factors, to provide a scientific basis for safeguarding the physical and mental well-being of adolescents.

### Definition of the Indicator

Cyberbullying can manifest in the following forms [39,40,41,42]: ① experiencing verbal insults, threats, and other forms of virtual violence in online spaces; ② being subjected to harassing messages, images, and other disruptive content in virtual environments; ③ having private photos or videos leaked online to humiliate, threaten, or mock the individual. In this study, the term “cyberbullying victimization” is defined as experiencing any of the aforementioned forms of cyberbullying at least once in the past year, without having engaged in any such behaviors towards others.

Perpetration of cyberbullying [39,40] can take the following forms: ① sending insulting language, threats, or other forms of virtual violence to others in online spaces; ② sending harassing messages, images, or other disruptive content to others in virtual environments; ③ leaking private photos or videos of the victim online to humiliate, threaten, or mock them. In this study, the term “cyberbullying perpetration” is defined as having engaged in any of the aforementioned forms of cyberbullying at least once in the past year, without being subjected to any such behaviors.

The term “ cyberbullying–victimization” is defined as experiencing and perpetrating cyberbullying in the past year, characterized by encountering any of the aforementioned forms of cyberbullying at least once and engaging in any of these behaviors at least once.

## 2. Materials and Methods

### 2.1. Subject of Investigation

From 1 March to 31 May 2019, there were 13 high schools and 33 junior high schools in Yixing, with a student distribution ratio of 2:1 between junior high and high school students. This study employed a random cluster sampling method, selecting 4 high schools from the 13 in Yixing and 3 junior high schools from the 33, resulting in a total of 7 schools and 13,258 students as study participants. A total of 13,258 questionnaires were distributed and, after excluding invalid responses, 12,770 valid questionnaires were obtained, yielding a response rate of 96.3%.

Prior to the study, informed consent was obtained from the participants and their parents or guardians. The research protocol was approved by the Ethics Review Committee of the Jiangsu Provincial Center for Disease Control and Prevention (Approval No.: JSJK2018-B025-02).

### 2.2. Method

The research team reviewed the primary manifestations of cyberbullying, referencing key studies, such as the China Education Tracking Survey (CEPS) project and the American Youth Risk Behavior Survey [43]. Additionally, a comprehensive literature review was conducted both domestically and internationally [44,45,46], and a self-administered questionnaire was developed. To ensure the reliability and validity of this questionnaire, it underwent rigorous evaluation, modification, and validation by relevant scholars, achieving high ratings throughout the process. Cronbach’s alpha was used to measure internal consistency, yielding a coefficient of 0.71. Content validity was assessed through correlation analysis, with coefficients ranging from 0.69 to 0.82. Prior to conducting the survey, the group provided specialized training for all relevant staff. During the survey, participants were informed about the study and participated with full consent. To address potential psychological issues among students, counselors were present in each class. Upon completion of the survey, data entry was conducted using a dual-entry system with two operators, and logical checks were performed to exclude any non-compliant questionnaires.

Participants mainly responded to the following questions regarding the frequency of different forms of cyberbullying they had experienced or perpetrated in the past 12 months. The responses were recorded on a scale from 1 to 5, where 1 represents “never”, 2 represents “once”, 3 represents “twice”, 4 represents “three times” and 5 represents “more than three times”.

General demographic characteristics: gender (male/female), ethnicity (Han/other), and educational level (middle school/high school).

Personal characteristics: smoking (yes/no), drinking (yes/no), serving as class monitor (yes/no), academic performance (excellent/above average/average/below average), preferred game type (board games/competitive games/strategy simulation games/role-playing games/do not play games), internet use time (<0.5 h/0.5–1 h/1–1.5 h/1.5–2 h/≥2 h), frequency of using social media (always/often/occasionally/never).

Family characteristics: family relationship (harmonious/average/occasional conflict/high conflict), parenting style (punishment and scolding/punishment and persuasion/scolding/persuasion/freedom), parental smoking (yes/no), parental drinking (yes/no), and whether both parents work outside of their hometown (yes/no).

School characteristics: peer acceptance (very popular/popular/somewhat unpopular/unpopular).

### 2.3. Statistical Analysis

Data entry was conducted using Epidata 3.0, with double entry by two individuals on separate machines to establish the database. Statistical descriptions of rates and composition ratios were performed using SPSS 25.0 software. The chi-square (χ^2^) test was applied to compare rates or composition ratios between groups, and a multivariate Logistic regression model was employed to analyze associated risk factors. The R software version 4.3.1 was used to create a nomogram prediction model, and the concordance index (C-index) was calculated to assess the model’s discrimination ability. Additionally, ROC curves and calibration curves were plotted to evaluate the predictive performance of the model. The significance level was set at α = 0.05.

## 3. Results

This study included a total of 12,770 middle and high school students, of whom 6550 were boys (51.3%) and 6220 were girls (48.7%). The cohort consisted of 8001 middle school students (62.7%) and 4769 high school students (37.3%). Among them, 7639 were only children (59.8%) and 5131 were non-only children (40.2%). Within this population, 10,723 students (84.0%) neither engaged in nor experienced cyberbullying, 1562 students (12.2%) were victims of cyberbullying only, 115 students (0.9%) were perpetrators of cyberbullying only, and 370 students (2.9%) both perpetrated and were victims of cyberbullying. Among those involved in both perpetrating and experiencing cyberbullying, 263 were boys (71.1%) and 107 were girls (28.9%); 213 were middle school students (57.6%) and 157 were high school students (42.4%).

As shown in Table 1, the prevalence of cyberbullying–victimization involvement was slightly higher among male students, those in higher grades, those who smoked or drank alcohol, those with below average academic performance, and those who were less popular among their peers (*p* < 0.05). In terms of media usage, adolescents who preferred playing single-player or multiplayer games, or engaging in role-playing games, exhibited a higher prevalence of cyberbullying–victimization involvement (*p* < 0.05). The prevalence of cyberbullying–victimization involvement also increased with longer internet usage and more frequent use of social networking or chat software (*p* < 0.05). Regarding family factors, adolescents from disharmonious families, those whose parents employed physical punishment or a combination of punishment and communication, those whose parents worked away from home, and those whose mothers smoked or drank alcohol had a relatively higher prevalence of cyberbullying–victimization (*p* < 0.05).

With adolescent cyberbully-victims as the dependent variable (0 = No; 1 = Yes), a logistic regression analysis was conducted using the following independent variables: gender (0 = Female; 1 = Male), school level (0 = Junior High; 1 = Senior High), personal alcohol consumption (0 = No; 1 = Yes), personal smoking (0 = No; 1 = Yes), academic performance (0 = Excellent; 1 = Above Average; 2 = Average; 3 = Below Average), level of peer popularity (0 = Very Popular; 1 = Quite Popular; 2 = Less Popular; 3 = Not Popular), preferred type of video game (0 = Never Play Video Games; 1 = Casual Games; 2 = Single or Multiplayer Battle Games; 3 = Simulation Strategy Games; 4 = Role-Playing Games), online time (h) (0~<0.5; 0.5~<1; 1~<1.5; 1.5~<2; ≥2), frequency of using social networking or chat apps (0 = Never; 1 = Always; 2 = Often; 3 = Occasionally), family relationship (0 = Harmonious; 1 = Average; 2 = Occasionally Conflictual; 3 = Highly Conflictual), parenting style (0 = Permissive; 1 = Persuasive; 2 = Punitive; 3 = Combined Punitive and Persuasive), father’s smoking status (0 = No; 1 = Yes), father’s alcohol consumption (0 = No; 1 = Yes), mother’s smoking status (0 = No; 1 = Yes), mother’s alcohol consumption (0 = No; 1 = Yes), and whether both parents work out of town (0 = No; 1 = Yes).

The results showed that factors associated with a higher risk of adolescent cyberbully-victims include being male, having both parents working out of town, having an average or occasionally conflictual family relationship, experiencing a punitive parenting style, personal smoking, personal alcohol consumption, always using social networking apps, and preferring single or multiplayer battle games. See Table 2.

A nomogram predictive model for adolescent cyberbully-victims was constructed using variables with statistically significant effect sizes in the multivariate analysis (*p* < 0.05).

As shown in Figure 1, the nomogram model was established based on the factors identified through multivariate analysis. Each value of a factor corresponds to a score on the first row of the nomogram labeled “Individual Score”. The scores for all factors are summed to yield a total score, which then corresponds to a probability of cyberbully-victims’ risk on the bottom row of the nomogram. After 1000 bootstrap internal validations, the model demonstrated a concordance index (C-index) of 0.82 (95% CI: 0.80–0.84), indicating good discriminatory ability.

As shown in Figure 2, the calibration curve indicates that the predicted probabilities of cyberbully-victims closely match the observed values, with an average absolute error of 0.01. In Figure 3, the black solid line represents the ROC curve of the predictive model, with an area under the curve (AUC) of 0.82 (95% CI: 0.80–0.84), demonstrating strong predictive capability.

## 4. Discussion

This study found that the prevalence of cyberbullying–victimization status among adolescents is 2.9%. This rate is slightly lower than the findings of Chang et al. [47], who reported an 11.2% prevalence among 2992 high school students in Taiwan over the past year, and Albert et al. [48], who reported a 12% prevalence among 502 students in three middle schools in the United States. However, our results are consistent with those of Chen Jiayi et al. [49], who found a 2.8% prevalence among 2931 adolescents in Western China, and Goebert et al. [50], who reported a 3% prevalence among 677 multi-ethnic high school students in Hawaii. Additionally, Bradshaw et al. [51] conducted a 2012 online survey of 24,620 adolescents across 52 high schools in Maryland, finding a 2.3% prevalence, while Jing Wang et al. [52] reported a 1.9% prevalence among 7313 adolescents in grades 6–10 in the United States, both of which are lower than the prevalence found in our study. The variation in prevalence rates of cyberbullying–victimization status may be attributed to differences in bullying prevalence, as well as varying socioeconomic and cultural backgrounds [34]. Furthermore, the use of different assessment scales [53] and variations in the definitions and concepts of “cyberbullying” [54] may also impact study results.

Regarding gender differences in cyberbully-victims’ status, this study indicates that males are more likely to be cyberbully-victims, which aligns with some previous research findings [55]. Adolescent males often exhibit impulsive behavior and poor self-control, making them more prone to retaliate with cyberbullying after experiencing it themselves; emotional outbursts may be a significant contributing factor [56,57]. Research [58] indicates that males are more likely to exhibit aggressive behaviors in social interactions, and the anonymity and lack of constraints in online environments may further amplify these aggressive tendencies. Males are also more inclined to participate in online gaming, which could increase their chances of becoming involved in cyberbullying. However, other studies [59,60] have suggested that females may be more likely to engage in and simultaneously experience cyberbullying. This could be due to females’ tendency to use indirect forms of bullying, which the nature of cyberbullying may facilitate [61]. Additionally, females are reportedly 2.5 times more likely to experience cyberbullying than males [62], possibly because females spend more time online sharing daily activities, thereby increasing their risk of cyberbullying.

This study also shows that excessive internet use and frequent use of social media platforms are significant risk factors for cyberbullying–victimization status among adolescents. The anonymity of the internet, along with the lack of eye contact, makes adolescents more likely to engage in self-disclosure and emotional expression, which may lead to increased aggression and impulsive behavior [63]. Adolescents who frequently use social media are more likely to encounter strangers, and this high level of online interaction increases their likelihood of becoming involved in cyberbullying incidents [47]. Compared to those who are only victims of cyberbullying, cyberbully-victims spend more time online and exhibit higher levels of depression, anxiety, and loneliness [64]. While the internet provides adolescents with opportunities to acquire new knowledge and make friends [65], it also poses negative impacts, such as addiction to violent games, exposure to Pornhub, and gambling [66,67].

Adolescents who frequently use social media are more likely to encounter strangers, which may increase the likelihood of cyberbullying incidents [45]. Behaviors such as smoking and drinking can label adolescents as “troublesome”, affecting their social relationships and reducing their popularity, which may lead them to seek comfort online, thereby increasing their risk of becoming cyberbully-victims [68]. To protect adolescents, schools and parents need to implement effective preventive measures to reduce these risky behaviors and pay attention to adolescents’ social interactions to lower their risk of becoming cyberbully-victims.

In this study, adolescents who preferred single or multiplayer battle games had a higher risk of cyberbully-victim status compared to those who preferred other types of games or did not play video games at all. This may be because such games often contain elements of violence and competition, and prolonged exposure may enhance adolescents’ aggressive tendencies, making them more likely to engage in bullying or victimization behaviors in the online environment [69,70]. Additionally, the interactive and anonymous nature of these games may foster unchecked behaviors, making adolescents more prone to becoming involved in online conflicts [71]. Cheng and Lin et al. [11] found a positive correlation between exposure to violent online games and cyberbully-victims’ behavior. James et al. [72] discovered that cyberbully-victims differ from pure bullies and victims in having higher levels of family conflict, proactive/reactive aggression, lower self-control, and weaker social bonds. This supports Schwartz’s [73] hypothesis that cyberbully-victims exhibit more proactive/reactive aggression. Targeted prevention and intervention strategies should be developed for this highly aggressive cyberbullying–victimization group.

The study also suggests that poor family relationships, punitive parenting styles, and parents working away from home may increase the risk of cyberbully-victim status among adolescents. These findings are similar to those of Sourander et al. [74]. Family environment significantly impacts adolescents’ mental health and social adaptation; a lack of family cohesion, poor communication between parents and children, and family conflicts indicate a certain level of family dysfunction, leading to feelings of neglect, loneliness, and emotional distress among adolescents [75,76]. These negative experiences may drive them to seek validation and engagement in online spaces, increasing their likelihood of involvement in cyberbullying–victimization. Additionally, when parents use punitive measures, such as physical punishment or scolding, children may learn from these behaviors. According to Social Learning Theory [77], adolescents are likely to imitate their parents’ actions, viewing violence as a way to solve problems, which increases the likelihood of them engaging in cyberbullying. Without effective parental supervision, adolescents may engage in risky behaviors such as meeting online acquaintances offline, watching inappropriate videos, or managing social media accounts, which can have serious consequences. Therefore, parents need to strengthen supervision and guidance to reduce the risk of cyberbullying–victimization [78].

There is also a correlation between cyberbullying–victimization status and academic performance. Research [79] shows that adolescents who experience cyberbullying have lower classroom attention, which negatively impacts their academic performance. Graham et al. [80] found that cyberbully-victims perceive school as an unsafe place and have the lowest academic performance among all groups. This may be due to the negative effects of cyberbullying on adolescents’ mental health and learning attitudes. First, cyberbully-victims experience higher levels of depression, anxiety, and stress [22,23,24], and these negative emotions can impair their cognition and attention, ultimately affecting academic performance. Second, bully-victims often lack a sense of safety and belonging at school, which leads to decreased motivation to learn. Therefore, the impact of cyberbullying–victimization on adolescents is multifaceted and warrants the close attention of parents, teachers, and related professionals.

Schools can offer courses to improve students’ awareness of cyber-bullying and their ability to cope. At the same time, teachers should pay attention to students’ mental health, and identify and intervene with students who are at risk of cyber-bullying. For parents, the need is to strengthen supervision and guidance of the child’s network use, more communication, and understanding of children’s network activities and social status. Instead of using negative parenting stylesm such as punishment, positive communication and support are needed to help the child develop healthy social relationships and self-control. For the psychologist, we should provide psychological support and counseling for cyber-bullying–bullied teenagers to help them deal with negative emotions, and to improve their self-efficacy and social adaptability. For the relevant policy makers, we need to formulate and improve the relevant laws and regulations to strengthen the supervision and punishment of cyber-bullying, promote the network platform to assume more social responsibility, and establish reporting and handling mechanisms to ensure the safety of young people’s networking.

This study has several limitations. As it is cross-sectional in nature, causal relationships cannot be established, and the findings should be interpreted as a basis for future research. The retrospective data collection may introduce recall bias, and the use of self-reported measures increases the risk of social desirability bias, which may influence the accuracy of the reported data. Future research should adopt longitudinal designs and incorporate multi-source data from teachers and parents to reduce self-report bias. It is important to explore the characteristics and mechanisms of cyberbullying across different cultural contexts to develop more targeted intervention strategies. Additionally, investigating the role of social media platforms, as well as the technological methods used for the prevention and intervention of cyberbullying, represents a significant area for further study.

## 5. Conclusions

In summary, this study involved 12,770 middle and high school students, providing an in-depth analysis of cyberbullying–victimization among adolescents, along with their associated factors. The results showed that male gender, smoking and drinking behaviors, frequency of social media use, and the quality of family relationships were significantly related to the risk of cyberbullying and victimization, indicating that these variables play a critical role in the occurrence of cyberbullying–victimization.

Based on these findings, we recommend implementing multi-level intervention strategies. First, schools should develop specialized cyberbullying education programs, particularly targeting high-risk groups, to enhance their awareness of potential risks in the online environment. Second, improving family communication and support is essential to create a healthier family environment, which could reduce adolescents’ vulnerability to cyberbullying. Additionally, we suggest establishing guidelines for social media use to encourage responsible and healthy internet usage, thereby reducing the incidence of cyberbullying.

This study not only provides important empirical evidence for understanding cyberbullying–victimization among adolescents but also offers theoretical support for the development of related policies and interventions. Future research should continue to explore other potential influencing factors and evaluate the effectiveness of interventions to promote the mental health and well-being of adolescents.

## Figures and Tables

**Figure 1 children-11-01193-f001:**
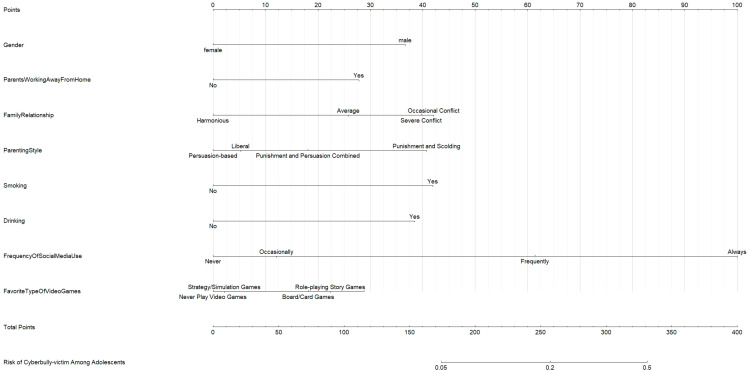
Nomogram Predictive Model for Adolescent Cyberbully-Victims’ Risk.

**Figure 2 children-11-01193-f002:**
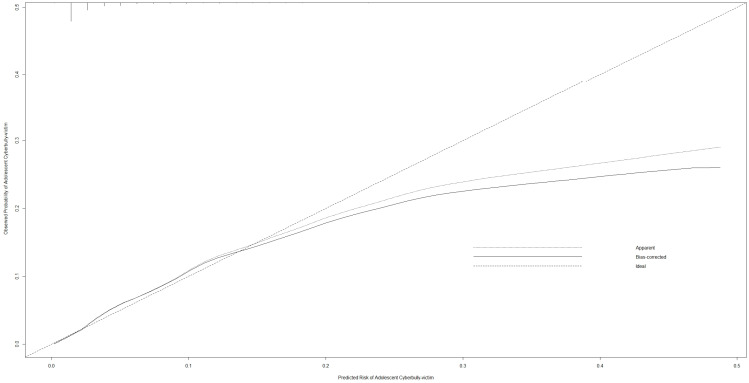
Calibration Curve for Adolescent Cyberbully-Victims’ Risk.

**Figure 3 children-11-01193-f003:**
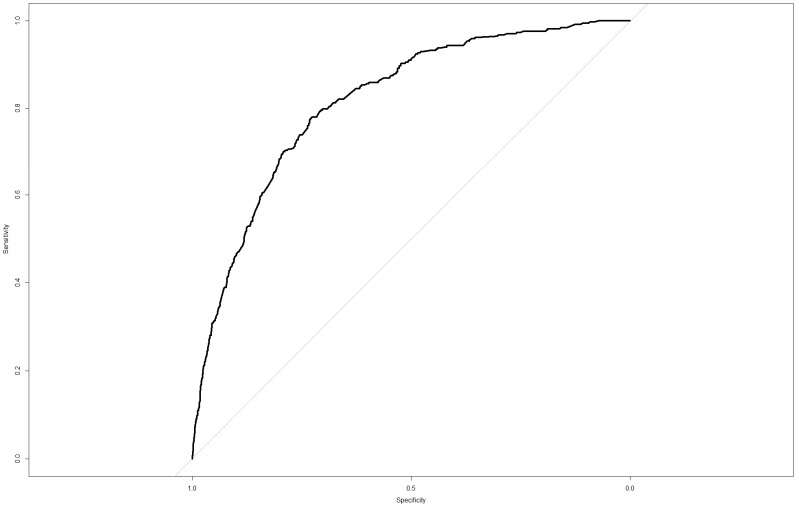
ROC Curve of the Nomogram Predictive Model for Adolescent Cyberbully-Victims’ Risk.

**Table 1 children-11-01193-t001:** Prevalence of cyberbullying–victimization Among Adolescents in Different Groups.

Category	Subgroup	Number	Cyberbullying–Victimization Number	χ^2^	*p*
Gender	Male	6550	263 (4.0)	59.727	<0.001
	Female	6220	107 (1.7)		
School level	Middle school	8001	213 (2.7)	4.214	0.040
	High school	4769	157 (3.3)		
smoking	Yes	365	58 (15.9)	225.457	<0.001
	No	12,405	312 (2.5)		
drinking	Yes	3110	203 (6.5)	192.542	<0.001
	No	9660	167 (1.7)		
Class monitor role	Yes	9819	285 (2.9)	0.004	0.950
	No	2951	85 (2.9)		
Academic performance	Excellent	1637	39 (2.4)	13.392	0.004
	Above Average	4097	118 (2.9)		
	Average	4174	103 (2.5)		
	Below Average	2862	110 (3.8)		
Popularity among peers *	Highly Popular	1926	61 (3.2)	22.306	<0.001
	Moderately Popular	9016	228 (2.5)		
	Less Popular	1385	60 (4.3)		
	Not Popular	264	15 (5.7)		
Favorite Type of Video Games	Board/Leisure Games	939	21 (2.2)	69.790	<0.001
	Single/Multiplayer Battle Games	6419	258 (4.0)		
	Strategy/Simulation Games	1282	22 (1.7)		
	Role-playing Story Games	1748	47 (2.7)		
	Never Play Video Games	2382	22 (0.9)		
Internet Usage Time (h)	0~<0.5	7218	138 (1.9)	131.156	<0.001 ^a^
	0.5~<1	2672	71 (2.7)		
	1~<1.5	1145	39 (3.4)		
	1.5~<2	656	36 (5.5)		
	≥2	1079	86 (8.0)		
Frequency of Social Media Use	Always	1855	159 (8.6)	245.766	<0.001 ^a^
	Often	4380	143 (3.3)		
	Occasionally	5535	60 (1.1)		
	Never	1000	7 (0.7)		
Only Child				1.237	0.266
	Yes	7639	211 (2.8)		
	No	5131	159 (3.1)		
Family Relationship *				113.804	<0.001
	Harmonious	8836	169 (1.9)		
	Average	1853	77 (4.2)		
	Occasional Conflicts	1682	96 (5.7)		
	Significant Conflicts	350	26 (7.4)		
Parenting Style *				64.061	<0.001
	Punishment-Based	347	30 (8.6)		
	Punishment and Persuasion Combined	3822	141 (3.7)		
	Liberal	2063	62 (3.0)		
	Persuasion-Based	6487	136 (2.1)		
Father’s Smoking Status				4.452	0.035
	Yes	7995	251 (3.1)		
	No	4775	119 (2.5)		
Father’s Drinking Status				5.392	0.020
	Yes	7997	253 (3.2)		
	No	4773	116 (2.4)		
Mother’s Smoking Status				14.247	<0.001
	Yes	149	12 (8.1)		
	No	12,621	358 (2.8)		
Mother’s Drinking Status				20.526	<0.001
	Yes	1497	71 (4.7)		
	No	11,273	299 (2.7)		
Parents Working Away from Home				20.724	<0.001
	Yes	1337	64 (4.9)		
	No	11,433	306 (2.7)		

^a^ linear trend chi-square test is used; the number in () is cyberbullying–victimization-reported rate (%); * indicates missing data.

**Table 2 children-11-01193-t002:** Multivariate Logistic Regression Analysis of Cyberbully-Victims Among Adolescents (n = 12,770).

Variable	Options	*β*	Standard Error	Wald Value	*p*	*OR* (95%CI)
Gender	female ^a^					1.00
	male	0.69	0.13	29.230	<0.001	2.00 (1.55~2.54)
Parents Working Away from Home	No ^a^					1.00
	yes	0.52	0.15	11.890	0.001	1.68 (1.25~2.25)
Family Relationship	Harmonious ^a^					1.00
	Average	0.44	0.15	8.698	0.003	1.55 (1.16~2.08)
	Occasional Conflicts	0.73	0.15	25.021	<0.001	2.07 (1.56~2.75)
	Significant Conflicts	0.62	0.25	6.424	0.011	1.86 (1.15~3.02)
Parenting Style	Liberal ^a^					1.00
	Persuasion-Based	−0.11	0.16	0.459	0.498	0.91 (0.65~1.23)
	Punishment-Based	0.56	0.26	4.623	0.032	1.75 (1.05~2.91)
	Punishment and Persuasion Combined	0.24	0.16	2.202	0.138	1.27 (0.93~1.76)
Drinking	no ^a^					1.00
	yes	0.71	0.12	35.353	<0.001	2.04 (1.61~2.58)
Smoking	no ^a^					1.00
	yes	0.84	0.18	23.032	<0.001	2.32 (1.64~3.27)
Frequency of Social Media Use	never ^a^					1.00
	Always	2.05	0.40	26.588	<0.001	7.77 (3.56~16.93)
	Often	1.35	0.40	11.553	0.001	3.84 (1.77~8.36)
	Occasionally	0.41	0.41	1.014	0.314	1.51 (0.68~3.33)
Favorite Type of Video Games	Never Play Video Games ^a^					1.00
	Board/Leisure Games	0.40	0.33	1.539	0.215	1.50 (0.79~2.84)
	Single/Multiplayer Battle Games	0.58	0.24	5.608	0.018	1.78 (1.10~2.87)
	Strategy/Simulation Games	0.07	0.32	0.046	0.830	1.07 (0.57~2.00)
	Role-playing Story Games	0.46	0.27	2.785	0.095	1.58 (0.92~2.71)

Note: Group ^a^ is the reference group.

## Data Availability

The data presented in this study are available on request from the corresponding author due to legal reason.

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
