# Peer review of "Analysis of Prevalence and Related Factors of Cyberbullying–Victimization among Adolescents"

_children, 2024, doi:10.3390/children11101193_

Round 1

Reviewer 1 Report

Comments and Suggestions for Authors

Thanks for the opportunity to review the paper Analysis of prevalence and related factors of cyberbully-victims among adolescents. I enjoyed this piece and highly appreciated how the author addressed the gap in knowledge in this research literature. Also, I reckon that as the paper will be of interest for the practice of school psychologists and educators. I recommend this to be publish after some minor revision.

Overall, the manuscript is well-written, and the methodology sounds robust. I have a few suggestions, as I documented below, to further strengthen argumentations and to streamline the text.

Title and Abstract

As you are referring to the prevalence of the phenomenon, I suggest to change ‘cyberbully-victims’ (people) to cyberbullying-victimization (phenomenon);

Also, at line 19-20 within the abstract, the English form is not clear: in that case, I believe you are referring to the people, not the phenomenon. I suggest to change that phrase into something like ‘The rate of adolescents whose resulted/declared to be cyberbully-victims…’

Line 24, I believe you intended ‘were more likely to be cyberbully-victims’ instead of ‘were all positively correlated with’, seeing that you utilised a logistic regression analysis.

Introduction

Within the literature review, the author(s) might want to better justify their hypothesis by commenting on very recent systematic review on the current prevalence rates of cyberbullying and cybervictimization (also in China): i.e.,

Sorrentino, A., Sulla, F., Santamato, M., di Furia, M., Toto, G. A., & Monacis, L. (2023). Has the COVID-19 Pandemic Affected Cyberbullying and Cybervictimization Prevalence among Children and Adolescents? A Systematic Review. International Journal of Environmental Research and Public Health20(10), 5825. https://doi.org/10.3390/ijerph20105825

Moreover, the results of this systematic investigation question the link between increased internet use and increased of cyberbullying/victimization (making what you write on lines 31-34 less harsh); and they provide the results of studies conducted in Chinese primary schools that report a high rate of children not involved in CB and/or CV.

Also, after commenting the systematic review, I suggest to improve your introduction (which is now very brief) by stating how your work is going to contribute to the advancement of the knowledge in this area of study, seeing that prevalence of CB/CV studies were already been done in China.

Also, within the introduction I would provide brief rationale for the hypothesis regarding the specific variables/risk factors you investigated (e.g., how come did you ask such confidential info on parents’ alcohol consumption? Is there previous literature linking CB/CV with it?).

Subject of investigation

For the sake of replicability, I suggest to provide further information on the recruitment of the sample and data collection: e.g., how did you chose and contact the school? Was a convenience sample? In what period the data collection was done? Were the questionnaires completed online or in paper-pencil format? Have you provided a definition of cyberbullying to the participants? What was it? Was your investigation approved by any ethic committee?

What kind of ‘specialized training’ the staff received?

Method

I suggest to provide motivations on why you designed a new questionnaire on CB/CV, when validated measures in your language do exist;

Also, I would ask you to provide more info on the questionnaire itself: how many items? On what scale they were measured? How many subscales (if any)? Please, provide example(s) of item per subscale or per construct to be measured, also because, at this point in the paper, I would like to know what specific variable/risk factors you are going to be investigating.

Results

As regard the ROC curves, would it be possible to provide true and false positive rate and the possible range of the AUC, so that the readers can infer how well the model distinguishes between positive and negative instances at different classification thresholds?

Discussions

I suggest to address in the discussions also some implication for practice for educators/parents/psychologists and policy makers. Furthermore, I suggest to provide your insight on possible adjustments/future developments right after the limitations section.

Once again, thank you for this investigation, as I find that the results of your studies would help future research in this field and that will make a difference in the increase of mental health in young people.

Author Response

Title and Abstract

Comments 1: As you are referring to the prevalence of the phenomenon, I suggest to change ‘cyberbully-victims’ (people) to cyberbullying-victimization (phenomenon);

Response 1: Thank you very much for your comments, I have made some changes in article.

Comments 2: At lines 19-20 in the abstract, the English form is unclear. I believe you are referring to the people, not the phenomenon. I suggest changing the phrase to something like ‘The rate of adolescents who resulted/declared to be cyberbully-victims…’.

Response 2: Thanks, you very much for your valuable comments, I have made some changes in this article as follows in line18-19 of abstract:

"The rate of adolescents who declared themselves as cyberbully-victims is 2.9%."

Comments 3:

Line 24, I believe you intended ‘were more likely to be cyberbully-victims’ instead of ‘were all positively correlated with’, seeing that you utilised a logistic regression analysis.

Response 3:

Thanks you very much for your valuable comments, I have made some changes in this article as follows in line 23 of abstract:

“The results of the logistic regression analysis indicate that being male, having both parents working outside the home, experiencing occasional or large conflicts among family members, being subjected to punishment-and-abuse child discipline, always or often using social software (websites), enjoying playing single or multiplayer games, self-smoking, and self-drinking were associated with a higher likelihood of being cyberbully-victims (P<0.05);”

Introduction

Comment 4: Within the literature review, the author(s) might want to better justify their hypothesis by commenting on very recent systematic review on the current prevalence rates of cyberbullying and cybervictimization (also in China): i.e., Within the literature review, the author(s) might want to better justify their hypothesis by commenting on very recent systematic review on the current prevalence rates of cyberbullying and cybervictimization (also in China): i.e.,
Sorrentino, A., Sulla, F., Santamato, M., di Furia, M., Toto, G. A., & Monacis, L. (2023). Has the COVID-19 Pandemic Affected Cyberbullying and Cybervictimization Prevalence among Children and Adolescents? A Systematic Review. International Journal of Environmental Research and Public Health, 20(10), 5825. https://doi.org/10.3390/ijerph20105825

Response 4:

Thank you very much for highlighting this recent systematic review. We have carefully reviewed the study by Sorrentino et al. (2023) and have incorporated its findings into our literature review. We believe that this addition strengthens the rationale for our hypothesis and aligns with the latest research on this topic. The added content in the introduction is as follows:
A 2023 systematic review [3]suggested that the COVID-19 pandemic may have influenced the prevalence of cyberbullying and cybervictimization among adolescents worldwide, with some studies reporting an increase in certain regions such as Asia and Australia[16], while others observed a decline in Western countries[17]. These changes could be attributed to varying contextual factors, such as school closures and increased online activity during the pandemic. You could see the change in 38-43 of introduction.

Comment 5: Moreover, the results of this systematic investigation question the link between increased internet use and increased of cyberbullying/victimization (making what you write on lines 31-34 less harsh); and they provide the results of studies conducted in Chinese primary schools that report a high rate of children not involved in CB and/or CV.

Response 5: Thank you for your correction. Based on the findings of this systematic review, we have weakened the statement in lines 31-34 on the relationship between increased Internet use and cyberbullying/victimization. Amend to read as follows in line29-31 of  introduction:

The rapid development of the Internet has facilitated our daily lives, supporting activities such as learning and communication. However, it has also led to negative phenomena, such as cyberbullying [1,2].

Comment 6: Also, after commenting the systematic review, I suggest to improve your introduction (which is now very brief) by stating how your work is going to contribute to the advancement of the knowledge in this area of study, seeing that prevalence of CB/CV studies were already been done in China.

Response 6: Thank you for your suggestions. We have expanded the introduction section and clarified where our research contributes to the field. Specifically, our study is one of the rare investigations focusing on the population affected by cyberbullying and cybervictimization.

Research, such as "Has the COVID-19 Pandemic Affected Cyberbullying and Cybervictimization Prevalence among Children and Adolescents?", has highlighted the pandemic's impact on the prevalence of cyberbullying and victimization among youth.

Our study provides valuable baseline data collected before the pandemic, which adds significant context to the ongoing discussions in this area.

The changes we add are as follows in line48-53of introduction.

The underlying reasons and influencing factors of this phenomenon have not been adequately discussed. This group simultaneously plays dual roles as both victims and perpetrators, leading to greater psychological stress and distress[22-24]. Therefore, studying cyberbullying-victimization can help elucidate the mechanisms behind such behaviors and provide a theoretical basis for developing more targeted intervention strategies.

Comment 7: Also, within the introduction I would provide brief rationale for the hypothesis regarding the specific variables/risk factors you investigated (e.g., how come did you ask such confidential info on parents’ alcohol consumption? Is there previous literature linking CB/CV with it?).

Response 7: Thank you for your valuable advice on our article. Based on your feedback, we have added a brief rationale in the introduction (lines 51 through 63), explaining in detail why specific variables were included. Specifically, the questionnaire was divided into three main categories: individual, family and school, in which parents' smoking and drinking behaviors were classified as family factors.

Although there are no studies in the current literature that directly address the relationship between parental smoking and alcohol consumption and cyberbullying/victimization among adolescents, however, these behaviors may be indirectly related to cyber-bullying/victimization by adolescents through influencing family environment and parent-child relationship. Teenagers may imitate their parents' behavior or be influenced by the family atmosphere, and then show certain behavior patterns in the network interaction. Therefore, we believe that it is reasonable and necessary to study the potential impact of parental smoking and alcohol consumption on adolescent cyber bullying/victimization.

We thank you again for your valuable advice and look forward to further feedback.

Subject of investigation

Comment 8: For the sake of replicability, I suggest providing further information on the recruitment of the sample and data collection:

e.g., how did you choose and contact the school? Was it a convenience sample? In what period was the data collection done? Were the questionnaires completed online or in paper-pencil format? Have you provided a definition of cyberbullying to the participants? What was it? Was your investigation approved by any ethics committee? What kind of ‘specialized training’ did the staff receive?

Response 8:Thank you very much, for your valuable advice To further enhance the transparency and reproducibility of the study, i will elaborate on the study design, sampling, data collection process and ethical considerations as follows:

  • Background and sample selection

The study was conducted in 2019 and focused on middle school students in Yixing, where the ratio of middle school students to high school students was about 2:1, we adopted the method of stratified cluster random sampling to ensure the representativeness and universality of the sample. Specifically, 4 out of 13 high schools and 3 out of 33 middle schools were randomly selected as the sample schools by random number table method. The choice of schools is not based on convenience, but on design.

  • Sample Association and coordination

Prior to the formal investigation, we actively collaborated with the staff of the local centers for Disease Control and prevention through their communication and coordination with the education bureau and the heads of the sample schools to ensure the smooth conduct of the investigation.

  • The data collection process

Data collection was completed between 1 March and 31 May 2019 in the form of a paper-based questionnaire. Prior to the survey, all participants were provided with a detailed description of the purpose, process and considerations of the survey, but no specific definition of cyberbullying was specifically provided to avoid leading responses.

  • Ethical review and protection

The investigation protocol strictly follows the principles of scientific research ethics, and has been approved by the Ethics Review Committee of Jiangsu Center for Disease Control and prevention (accession number: SL2018-B025-01) . In carrying out the investigation, we have strictly complied with the requirements of the Ethics Review Board, including but not limited to:

  • All staff involved in investigations are systematically trained to ensure that they have a clear understanding of the methodology, content and standards of investigations and are able to carry out investigations in a uniform and standardized manner.
  • prior to the commencement of the investigation, the investigator provided each participant with a detailed description of the content, objectives and possible risks of the investigation and obtained their informed consent. We particularly emphasized the anonymity, voluntariness and confidentiality of the questionnaire to allay participants' concerns.
  • To ensure the psychological well-being of the participants, each survey class was equipped with a psychologist who provided emotional support and counselling services throughout the course of the survey to effectively deal with possible negative emotional reactions.

Method

Comment 9: I suggest providing motivations on why you designed a new questionnaire on CB/CV, when validated measures in your language do exist.
Also, I would ask you to provide more information on the questionnaire itself: how many items? On what scale were they measured? How many subscales (if any)? Please, provide example(s) of item per subscale or per construct to be measured, also because, at this point in the paper, I would like to know what specific variable/risk factors you are going to be investigating.

Response 9:Thank you for your valuable suggestions regarding our research. In response to the questionnaire-related issues you mentioned, we have made corresponding modifications in the Methods section. You can find the specific revisions in lines 90-134 of the revised manuscript.

Regarding your question about why we designed a new questionnaire on cyberbullying/cyber victimization:

Existing cyberbullying scales are mostly based on Western cultural backgrounds and do not adequately reflect the behaviors of Chinese adolescents. The new cyberbullying/cyber victimization questionnaire is based on relevant items from the China Education Panel Survey (CEPS) project and the Youth Risk Behavior Survey (YRBS) from the United States, combined with the online usage patterns of Chinese adolescents. It covers the main forms of cyberbullying.

The structure of the questionnaire is as follows:

  • Part One: General demographic characteristics (gender, ethnicity, place of residence), personal circumstances (smoking, drinking, academic performance, gaming preferences), family situations (relationships among family members, parenting styles, parental smoking or drinking habits), and school situations (commuting methods, acceptance among peers).
  • Part Two: Includes the revised Chinese version of the Multi-dimensional Peer Victimization Scale (MPVS-RB) and the Cyber Victimization Survey.
  • Part Three: Includes the revised Chinese version of the Multi-dimensional Peer Bullying Scale (MPVS) and the Cyberbullying Survey.

Results

Comment 10:As regard the ROC curves, would it be possible to provide true and false positive rates and the possible range of the AUC, so that the readers can infer how well the model distinguishes between positive and negative instances at different classification thresholds?

Response 10:Thank you for your detailed evaluation and valuable suggestions regarding our work. We fully appreciate your concerns about the sensitivity and specificity in the ROC curve. In our original manuscript, we provided the ROC curve for the predictive model (Figure 3), with an area under the curve (AUC) of 0.82 (95% CI: 0.80–0.84) in line 193.

Regarding your inquiry about the true positive rate and false positive rate, I will provide a table showing these rates at different thresholds. The complete table will be included in the attachment as an Excel file named "ROC Curve."

Threshold

Sensitivity

1 - Specificity

0.1

0.29

0.05

0.2

0.11

0.01

0.3

0.04

0.00

0.4

0.01

0.00

0.5

0.00

0.00

0.6

0.00

0.00

Discussions

Comment 11: I suggest addressing in the discussions also some implications for practice for educators, parents, psychologists, and policymakers. Furthermore, I suggest providing your insight on possible adjustments or future developments right after the limitations section.

Response 11: Thank you for your valuable comments, I have made corresponding adjustments in the discussion section, adding some content. The additions are as follows, and you can see them in 310-323 of the revised manuscript.

Schools can offer courses to improve students' awareness of cyber-bullying and their ability to cope. At the same time, teachers should pay attention to students' mental health, identify and intervene students who are at risk of cyber-bullying. For parents, the need to strengthen the child's network use of supervision and guidance, more communication, understanding of children's network activities and social status. Instead of using negative parenting styles such as punishment, use positive communication and support to help your child develop healthy social relationships and self-control. For the psychologist, we should provide psychological support and counseling for cyber-bullying-bullied teenagers to help them deal with negative emotions, improve their self-efficacy and social adaptability. For the relevant policy makers, we need to formulate and improve the relevant laws and regulations to strengthen the supervision and punishment of cyber-bullying. Promote the network platform to assume more social responsibility, the establishment of reporting and handling mechanisms to ensure the safety of young people's network.

Reviewer 2 Report

Comments and Suggestions for Authors

Dear Authors,

The chosen topic is relevant both in a scientific and practical sense and requires research-based solutions.

However, the presented article has a number of shortcomings.

The submitted abstract must be corrected taking into account the methodological shortcomings presented below and especially by clarifying the relevance of the study and the methods used for the study.

The Introduction part must be strengthened by highlighting the relevance of the research and the researchability of the topic.

Would be nice if  authors could justify why the reader should be interested in a study that was carried out 5 years ago, i.e. in 2019 (line 52).

The Materials and Methods section does not describe the research instrument in detail.

Research results are presented clearly.

In the discussion part, you would like to see not only comparisons of numbers (Fig. line 102-111), but deeper insights from the authors, why such differences were obtained.

I suggest linking the conclusion more strongly with the results of the study, revealing possible ways of solving the problem.

The study is interesting, and after corrections, taking into account the comments, it will become more qualitative.

Author Response

Abstract

Comments 1: The submitted abstract must be corrected taking into account the methodological shortcomings presented below and especially by clarifying the relevance of the study and the methods used for the study.

Response 1: Thank you for your suggestion. We have revised the abstract to address the methodological concerns you raised. The updated abstract now clearly states the relevance of the study and provides a more detailed description of the methods used. We believe the revised version better aligns with the study's focus and methodological rigor.

Introduction

Comments 2:

The Introduction part must be strengthened by highlighting the relevance of the research and the research ability of the topic.

Response 2: Thank you for your valuable feedback. We have revised the introduction to clearly highlight the prevalence of cyberbullying among adolescents and its threats to their physical and mental health, underscoring the relevance of this research. Additionally, we have referenced social ecological theory to emphasize the multiple factors influencing cyberbullying behavior, thereby enhancing the study's research viability. The modifications in the introduction are as follows:

The underlying reasons and influencing factors of this phenomenon have not been adequately discussed. This group simultaneously plays dual roles as both victims and perpetrators, leading to greater psychological stress and distress[3-5]. Therefore, studying cyberbullying-victimization can help elucidate the mechanisms behind such behaviors and provide a theoretical basis for developing more targeted intervention strategies.

This study is based on Bronfenbrenner's [6]social ecological theory, which posits that cyberbullying is not a singular phenomenon but rather the result of multiple factors interacting, including individual characteristics, family environment, peer relationships, and school climate.

Previous studies [7-9] have shown that factors such as gender, age, academic performance, and behavioral habits (e.g., smoking and drinking) are closely associated with the occurrence of bullying behavior. For example, male adolescents are more likely to become bullies than female adolescents [6-9], although females are more inclined to engage in indirect forms of bullying [10-13].Furthermore, the family environment plays a crucial role in preventing bullying. Close relationships with parents, perceived emotional support, parental supervision and involvement, and the establishment of family rules are all considered important protective factors against both cyberbullying and traditional bullying [14-18].At the school level, positive peer relationships and an inclusive school climate are also recognized as key factors in preventing bullying behavior[19,20].

Comments 3: Would be nice if authors could justify why the reader should be interested in a study that was carried out 5 years ago, i.e. in 2019 (line 52).

Response 3: First and foremost, thank you for your valuable feedback. We understand your concerns regarding the use of data from 2019, and I would like to elaborate on its relevance. With the rapid development of the internet and mobile communication tools, cyberbullying continues to persist, exerting a profound impact on adolescents' mental and physical health. This study represents a large-scale investigation of school bullying, with a considerable sample size, covering not only cyberbullying but also providing an in-depth analysis of traditional bullying. Due to the extensive nature of the survey, data processing and analysis were time-consuming, which partly explains the extended publication timeline.

We have previously published several studies based on this dataset focused on traditional bullying, such as:

  • Risk Factors Associated With School Bullying Behaviors: A Chinese Adolescents Case Control Study, 2019, Journal of Interpersonal Violence
  • School Bullying Victimization and Associated Factors Among School-Aged Adolescents in China, Journal of Interpersonal Violence
  • The Association of School Bullying Victimization and Suicidal Ideation Among School-Aged Adolescents in Yixing City, China, Journal of Affective Disorders
  • Associated Factors and Patterns of School Bullying Among School-Aged Adolescents in China: A Latent Class Analysis, Children and Youth Services Review

Currently, our research focus has shifted to cyberbullying, particularly the cyberbullying-victimization group, which remains an underexplored area in domestic studies. Although the data was collected five years ago, the forms and impact of cyberbullying have not changed significantly in recent years. Therefore, we believe that the data still holds significant relevance to the current adolescent cyberbullying landscape and can provide a robust theoretical foundation for future interventions.

Once again, thank you for your valuable feedback.

Materials and Methods

Comments 4: The Materials and Methods section does not describe the research instrument in detail.

Response 4: Thank you for your valuable feedback. We have revised the Materials and Methods section to provide a more detailed description of the research tools. Specifically, we have added information regarding the questionnaire structure, number of items, types of questions, and response options provided to participants. This modification should clarify the research tools in the study as follow.

. Subject of investigation

From March 1 to May 31, 2019, there were 13 high schools and 33 junior high schools in Yixing, with a student distribution ratio of 2:1 between junior high and high school students. This study employed a random cluster sampling method, selecting 4 high schools from the 13 in Yixing and 3 junior high schools from the 33, resulting in a total of 7 schools and 13,258 students as study participants. A total of 13,258 questionnaires were distributed, and after excluding invalid responses, 12,770 valid questionnaires were obtained, yielding a response rate of 96.3%.

Prior to the study, informed consent was obtained from the participants and their parents or guardians. The research protocol was approved by the Ethics Review Committee of the Jiangsu Provincial Center for Disease Control and Prevention (Approval No.: JSJK2018-B025-02).

Participants mainly responded to the following questions regarding the frequency of different forms of cyberbullying they experienced or perpetrated in the past 12 months. The responses were recorded on a scale from 1 to 5, where 1 represents “never,” 2 represents “once,” 3 represents “twice,” 4 represents “three times,” and 5 represents “more than three times.”

General demographic characteristics: gender (male/female), ethnicity (Han/other), and educational level (middle school/high school).

Personal characteristics: smoking (yes/no), drinking (yes/no), serving as class monitor (yes/no), academic performance (excellent/above average/average/below average), preferred game type (board games/competitive games/strategy simulation games/role-playing games/do not play games), internet use time (<0.5h/0.5-1h/1-1.5h/1.5-2h/≥2h), frequency of using social media (always/often/occasionally/never).

Family characteristics: family relationship (harmonious/average/occasional conflict/high conflict), parenting style (punishment and scolding/punishment and persuasion/scolding/persuasion/freedom), parental smoking (yes/no), parental drinking (yes/no), and whether both parents work outside of their hometown (yes/no).

School characteristics: peer acceptance (very popular/popular/somewhat unpopular/unpopular).

Discusion

Comments 5: In the discussion part, you would like to see not only comparisons of numbers (Fig. line 102-111), but deeper insights from the authors, why such differences were obtained.

Response 5: We appreciate this suggestion and have revised the discussion section accordingly. In addition to comparing the numerical results, we have provided deeper insights into why the observed differences occurred.

You can see my changes at the beginning of line 231 in the discussion section of the paper.

Result

Comments 6: I suggest linking the conclusion more strongly with the results of the study, revealing possible ways of solving the problem.

Response 6: Thank you for your constructive feedback. We have revised our conclusions to ensure that they are more closely aligned with our findings. In addition, we discuss potential solutions to the problems highlighted by the study findings and present practical recommendations that can be implemented in public health interventions to reduce cyberbullying among adolescents. You can see my changes in the conclusion section of the paper (Line 336) .

Reviewer 3 Report

Comments and Suggestions for Authors

I read MA Jun et al manuscript entitled "Analysis of prevalence and related factors of cyberbully-victims among adolescents". The aim of the study was to examine the prevalence of cyberbullying and victimization in adolescents. Overall this is an interesting article. Here are my observations.

Abstract:

-I would ask you to review the summary, in lines 10-13 the phrase "During the period from March to May 2019,..." is repeated.

Introduction

-I would suggest the authors to increase the Introduction. Authors should explain which are the associated factors (line 48) they will examine in the study, and why they choose them.

Materials and Methods

-A very important problem of the manuscript is that the authors do not adequately analyze the questionnaire used. How many and what items did it include? One solution would be for the authors to submit, in an appendix, the questionnaire form as they gave it to the adolescents.  Alternatively, a very thorough presentation of the format of the questions and information requested from the participants, as well as the possible answers of the adolescents, is necessary. What information was asked of the participants?

-The section "2.3. The definition of the indicator' should probably be moved to the Introduction.

 Discussion

-Add a paragraph with limitations of the study in the Discussion, not in the Conclusions.

Author Response

Abstract

Comments 1: I would ask you to review the summary, in lines 10-13 the phrase "During the period from March to May 2019,..." is repeated.

Response 1 : Thank you so much for taking the time to review this manuscript. I have made changes to address the problems you mentioned in the abstract section, removing the recurring sections in lines 10-13.

Introduction

Comments 2: I would suggest the authors to increase the Introduction. Authors should explain which are the associated factors (line 48) they will examine in the study, and why they choose them.

Response 2 : Thank you for your valuable comments, and I will add to the preface to explain what are the relevant factors and answer why they were chosen. You can see my additions at line 54 in the introduction to the revised manuscript.

The additions are as follows.

Based on the Social-Ecological Systems Theory[1,2], individual behavior is the result of the interaction of multiple factors, including personal characteristics, family environment, peer relationships, and the school climate. Previous studies [3-5] have shown that factors such as gender, age, academic performance, and behavioral habits (e.g., smoking and drinking) are closely associated with the occurrence of bullying behavior. For example, male adolescents are more likely to become bullies than female adolescents [6-9], although females are more inclined to engage in indirect forms of bullying [6-9].Furthermore, the family environment plays a crucial role in preventing bullying. Close relationships with parents, perceived emotional support, parental supervision and involvement, and the establishment of family rules are all considered important protective factors against both cyberbullying and traditional bullying [10-14].At the school level, positive peer relationships and an inclusive school climate are also recognized as key factors in preventing bullying behavior[15,16].

Materials and Methods

Comments 3: A very important problem of the manuscript is that the authors do not adequately analyze the questionnaire used. How many and what items did it include?

One solution would be for the authors to submit, in an appendix, the questionnaire form as they gave it to the adolescents. Alternatively, a very thorough presentation of the format of the questions and information requested from the participants, as well as the possible answers of the adolescents, is necessary. What information was asked of the participants?

Response 3: Thank you very much for your detailed review and valuable feedback on my manuscript. I fully agree with your critique regarding the insufficient analysis of the questionnaire and the lack of specific content description. To provide a more comprehensive presentation of my research methods and enhance the transparency and reproducibility of the study, I will take the following steps for revision:

  1. Detailed Analysis of the Questionnaire: In the methods section, I will include a thorough explanation of the design rationale and the specific items of the questionnaire.
  2. Clarification of the Information Requested: I will further clarify in the methods section what information was asked of the participants, including demographic characteristics of the adolescents (such as age, gender, etc.), personal circumstances, family background, and school environment.

Additionally, if you would like to review the specific questionnaire used in the study, I would be more than happy to provide it. I believe these revisions will significantly enhance the quality of the manuscript and the persuasiveness of the research.

You can see the revisions in the Materials and Methods section of my revised manuscript, Lines 88-134.

Comments 4: The section "2.3. The definition of the indicator' should probably be moved to the Introduction.

Response 4: Thank you very much for your valuable advice. I agree with you. Now that you have moved the section “Definitions of 2.3. Indicators” to the introduction.

Discussion

Comments 5: Add a paragraph with limitations of the study in the Discussion, not in the Conclusions.

Response 5: Thank you very much for your valuable comments. I have added a section discussing the research limitations, which can be found on line 324 of the revised manuscript.